# Oncometabolites—A Link between Cancer Cells and Tumor Microenvironment

**DOI:** 10.3390/biology11020270

**Published:** 2022-02-09

**Authors:** Maksymilian Baryła, Aleksandra Semeniuk-Wojtaś, Letycja Róg, Leszek Kraj, Maciej Małyszko, Rafał Stec

**Affiliations:** 1Department of Oncology, Medical University of Warsaw, 02-097 Warsaw, Poland; maksdh2@interia.pl (M.B.); letycja.rog96@gmail.com (L.R.); leszekkraj@gmail.com (L.K.); maciekmalyszko@gmail.com (M.M.); drrafals@wp.pl (R.S.); 2Department of Molecular Biology, Institute of Genetics and Animal Biotechnology, Polish Academy of Sciences, 05-552 Jastrzębiec, Poland

**Keywords:** oncometabolite, lactate, fumarate, glutamate, succinate

## Abstract

**Simple Summary:**

Lactate, glutamate, fumarate, and succinate are metabolites that accumulate in tumors as a consequence of an alteration in cellular respiration connected with malignant transformation. These metabolites link all types of cells involved in tumor survival and progression, so they are also called oncometabolites. Here, we describe the pathways that lead to the accumulation of lactate, glutamate, fumarate, and succinate in solid tumors and their impact on shaping the tumor microenvironment. The data show that oncometabolites play a particularly important role in neoangiogenesis and in the regulation of the immune component of tumor. Oncometabolites are also associated with a disrupted DNA damage response and make the tumor microenvironment more favorable for cell migration. The knowledge summarized in this article will allow for a better understanding of the associations between cancer cells and the tumor microenvironment as well as the direct effects of these particles on cancer development.

**Abstract:**

The tumor microenvironment is the space between healthy tissues and cancer cells, created by the extracellular matrix, blood vessels, infiltrating cells such as immune cells, and cancer-associated fibroblasts. These components constantly interact and influence each other, enabling cancer cells to survive and develop in the host organism. Accumulated intermediate metabolites favoring dysregulation and compensatory responses in the cell, called oncometabolites, provide a method of communication between cells and might also play a role in cancer growth. Here, we describe the changes in metabolic pathways that lead to accumulation of intermediate metabolites: lactate, glutamate, fumarate, and succinate in the tumor and their impact on the tumor microenvironment. These oncometabolites are not only waste products, but also link all types of cells involved in tumor survival and progression. Oncometabolites play a particularly important role in neoangiogenesis and in the infiltration of immune cells in cancer. Oncometabolites are also associated with a disrupted DNA damage response and make the tumor microenvironment more favorable for cell migration. The knowledge summarized in this article will allow for a better understanding of associations between therapeutic targets and oncometabolites, as well as the direct effects of these particles on the formation of the tumor microenvironment. In the future, targeting oncometabolites could improve treatment standards or represent a novel method for fighting cancer.

## 1. Introduction

To survive and grow, cancer cells require an appropriate microenvironment that connects them with normal tissue, from which they can receive nutrients. The microenvironment also provides suitable conditions for tumor development, i.e., a low oxygen concentration, a strongly acidic environment, and suppression of the host’s immune system. The microenvironment includes immune cells, cancer-associated cells such as cancer-associated fibroblasts (CAFs), blood vessels, and stroma [1]. Another necessary condition for cancer development is an alteration in glucose metabolism. Several proteins are involved in the metabolic switch towards an anaerobic pathway and a key player in this process is hypoxia-induced factor 1α (HIF-1α), which is controlled by cellular oxygen concentrations. Hypoxic conditions prevent HIF-1α from being degraded in proteasomes via the von Hippel Lindau protein-mediated pathway, leading to its accumulation in cancer cells and its excretion into tumor stroma [2].

HIF-1α has a significant impact on angiogenesis, plays an important role in cell function, and promotes the adaptation of cancer cells to the conversion from an aerobic to anaerobic metabolism. HIF-1α target genes include genes that encode proteins involved in angiogenesis such as vascular endothelial growth factor (*VEGF*), platelet-derived growth factor (*PDGF*), placental growth factor (*PLGF*), and cyclooxygenase-2 (*COX-2*), as well as genes that participate in the epithelial-to-mesenchymal transition by hepatocyte growth factor receptor *(HGFR),* and in extracellular matrix formation by fibronectin, hydroxylated collagen IV and by integrin. HIF-1α is also connected with chemotaxis, cell proliferation, and survival by transforming growth factor-α (*TGF-α*) or epidermal growth factor receptor (*EGFR*) [3].

The formation of anaerobic cell metabolism is connected with the indirect stabilization of transcription factors and direct activation of genes involved in anaerobic metabolism, as well as in the suppression of enzymes of the tricarboxylic acid (TCA) cycle via HIF-1α. During an anaerobic metabolism only two adenosine triphosphate (ATP) molecules are produced from one glucose molecule via the anaerobic pathway, instead of the 36 ATP molecules yielded from aerobic respiration. This could be considered unfavorable for rapidly proliferating cells. Nevertheless, cancer cells also prefer anaerobic glycolysis under aerobic conditions because the anaerobic glucose metabolism supplies a lot of the substrates involved in the synthesis of nucleotides and amino acids, which are crucial for rapidly dividing cancer cells [4,5].

The microenvironment also includes a group of substances called oncometabolites such as lactate, fumarate, glutamate, and succinate, which are produced by tumor cells and other cells that participate in shaping the tumor microenvironment, such as CAFs and tumor-associated macrophages (TAMs). Under normal conditions, metabolites are located in the cell and are essential for ATP generation. In cancer cells with a disturbed energy metabolism, intermediate metabolites accumulate and can be detected not only in cells, but also in the extracellular matrix. The accumulated metabolites cause metabolic and non-metabolic dysregulation and compensatory responses in the cell, favoring malignant transformation [6]. They are, therefore, termed oncometabolites, but their role in the tumor remains unclear. There are also data showing that lactate overproduction in tumor can cause serum lactate concentration growth, which could be a predictor of unfavorable outcome in cancer patients [7,8]. The evidence, mainly derived in vitro and from animal models, as summarized below, indicates that oncometabolites provide a method of communication between the cancer cells and tumor-infiltrating cells that drive cancer progression. Here, we describe the changes in metabolic pathways that lead to the accumulation of lactate, glutamate, fumarate, and succinate in the solid tumors and their impact on shaping the tumor microenvironment. A closer understanding of the relationship between cancer cells and the tumor microenvironment is particularly important, as targeting oncometabolites could help to improve current treatment standards or represent a novel method for fighting against cancer [9,10]. Investigators also assess oncometabolites as a potential source of prognostic information and biomarkers of therapeutic efficacy; therefore, knowledge about the consequences of metabolic alteration is important from a clinical perspective [11,12].

## 2. Lactate

The malignant transformation of cancer cells is connected with an enhanced glucose metabolism. To ensure adequate ATP production, cells increase the expression of glycolytic enzymes and metabolite transporters, which is associated with increased lactate production. There are also some defects in oncogenes, which affect cancer cell metabolism. This includes, for example, the mutated *p53 gene,* associated with a decrease in *T*P53-*i*nduced *g*lycolysis and *a*poptosis *r*egulator (TIGAR), which normally inhibits glycolysis, and the expression and activation of glycolysis [13,14]. Moreover, newly transformed cells with a highly oxidative metabolism, in alternate stages under hypoxia, shift their metabolic profile to a glycolytic phenotype [15]. This phenomenon was observed for embryonic mouse cells after oncogenic transformation by Harvey rat sarcoma viral oncogene homolog (HRasV12) and adenovirus early region 1A (EA1).

To ensure adequate ATP production, cells increase the expression of glycolytic enzymes and metabolite transporters, which is associated with increased lactate production. Glutaminolysis is another pathway responsible for the production of energy and macromolecules that is associated with lactate overproduction.

### 2.1. Effects of Lactate on Angiogenesis

The lactate produced in glycolysis has an important effect on cancer development because it directly and indirectly stimulates neoplastic angiogenesis (Figure 1). Lactate molecules are thought to activate the nuclear factor kappa B/interleukin 8 (NF-kB/IL-8) pathway in endothelial cells, leading to formation of the chemokine CXCL-8, which is necessary for endothelial cell migration and plays a role in the formation of new vessels [16]. The expression of the lactate dehydrogenase 5 (LDH5) isoform, confirmed by immunohistochemistry (IHC), correlates with that of HIF-1α/2α and VEGF, as well as with vascular density in neoplastic tumors [17]. Lactate also stimulates angiogenesis in human tumor cells by preventing the inhibition of prolyl hydroxylase domain proteins (PHDs), resulting in the hydroxylation and stabilization of IκB kinase β (IKKβ) and HIF-1α. HIF-1α stabilization leads to the activation of HIF-1α-mediated VEGF signaling and the disturbance of IKKβ hydroxylation, resulting in activation of the NF-κB pathway [18,19]. Lactate can also contribute to angiogenesis by preventing the proteasomal degradation of the N-Myc downstream-regulated protein (NDRG3) in cancer cells lines, as well as in epithelial and fibroblast cell lines, which activates Raf/extracellular-signal-related kinase (ERK) signaling, promoting angiogenesis and cell growth [20].

Another mechanism by which lactate induces tumor angiogenesis is more complex and involves TAMs. TAMs have monocarboxylic acid transporter 1 (MCT1) on their cell membrane, which lactate molecules use to penetrate macrophages [21], where they are used in several separate pathways, leading to a pro-angiogenic phenotype [22]. The first of these mechanisms is based on the re-conversion of lactate to pyruvate, which results, among other things, in impaired HIF-1α ubiquitination and its accumulation in the cancer cells [23], as well as increased VEGF production, leading to an intensification of angiogenesis. The indirect stabilization of HIF-1α by lactate also contributes to activation of the carbonic anhydrase 9 (*CA9*) gene [24], encoding the transmembrane protein of the family of carbonic anhydrases (CAIX) [25], which maintains a constant acidic pH in the tumor tissue. Moreover, lactate probably stabilizes HIF-2α inside TAMs by activating the mammalian target of rapamycin complex 1 (mTORC1), which acts as a nutrient/energy/redox sensor and controls protein synthesis, leading to inhibition of the expression of transcription factor EB in xenograft mouse tumor models [26] and, subsequently, to the decreased expression of *Atp6v0d2*, which encodes the macrophage-specific ATPase subunit involved in HIF-2α degradation. When the amount of ATPase subunit is insufficient, HIF-2α accumulates in the cell, which stimulates the production of VEGF in TAMs. However, these data were obtained from murine models and require validation in human TAMs.

Angiogenesis is also connected with lactate’s ability to directly stimulate a pathway involving ERK1/2 and signal transducer and activator of transcription (STAT3) inside TAMs [27], as observed in breast cancer. In in vitro studies, enhanced proliferation, migration, and angiogenesis were reported in breast cancer cells when lactate-treated TAMs were added to cells cultures. In further steps of the experiments, activated ERK and STAT led to the polarization of monocytes towards the pro-angiogenic phenotype of TAMs. After the administration of selumetinib and static, an MEK and a STAT3 inhibitor, a decrease was observed in the expression of markers of lactate-activated TAMs, such as CD206 and arginase-1. Administration of the same inhibitors to TAMs incubated with lactate and breast cancer cells resulted in a reduction in in vitro proliferation, migration, and tumor angiogenesis.

In 2016, Mathias et al. found, in an animal model, that, under hypoxic conditions, TAMs exhibit an increased expression of regulated in development and DNA damage responses 1 (REDD1), a negative regulator of the mTOR pathway [28]. Compared to macrophages carrying the *Redd1* gene, the addition of macrophages lacking *Redd1* to Lewis lung carcinoma cells resulted in a smoothing of the endothelium and a reduction in the diameter of newly formed vessels. Similar results were obtained for breast cancer cells, in which the addition of *Redd1*-deficient TAMs was associated with the formation of vessels with more stable cell connections and higher pericyte coverage compared with TAMs carrying the *Redd1* gene [28]. This correlation shows that the presence of *Redd1* and its product in TAMs under conditions of increased glycolysis is associated with abnormal tumor angiogenesis.

Studies have also analyzed the effect of the acidic environment and lactate itself on the activation of transmembrane G proteins expressed on macrophages and the related stimulation of angiogenesis. However, researchers have not been able to clearly define the role of lactate in angiogenesis promotion via this pathway. Table 1 shows the lactate-modified pathways found in the tumor microenvironment and tumor-associated cells, especially in endothelial cells, neutrophils and TAMs.

### 2.2. Immunosuppressive Effect of the Acidic Environment and Lactate

A low pH in the microenvironment associated with increased lactate levels is also important in the context of the immune system (Figure 2). Lactate in the tumor microenvironment has an important influence on TAM polarization towards the immunosuppressive M2 phenotype detected in murine tumor models of lung carcinoma, melanoma and prostate cancer [29,30]. TAMs incubated in an acidic environment produced experimentally by adding lactate were observed to increase lymphocyte apoptosis when added to a T cell population, probably via interaction between programmed cell death 1 (PD1) and its ligand (PD-L1). Lactate-producing cells contributed to the increased expression of PD-L1 by TAMs and increased apoptosis of T cells compared to the control group treating without lactic acid, thus helping tumor cells to escape the immune system [24]. Although this was observed in pancreatic cancer, we cannot exclude the possibility that a similar association also takes place in renal cell cancer. In addition to effects on neovascularization, VEGF, the production of which is stimulated by HIF-1α and HIF-2α in macrophages, also affects the immune system. Investigators analyzed tumor cell lines and found that the presence of VEGF in the tumor microenvironment contributes to immune escape by cancer cells by preventing the differentiation and maturation of dendritic cells, thus reducing antigen presentation [31,32]. Besides, VEGF also could increase the expression of VEGFR in DCs [33]. VEGF also increases PD-L1 expression on the surface of dendritic cells [34], which leads to a reduction in T-cell cytotoxicity as a result of apoptosis via interactions between PD-L1 and PD-1. Additionally, in animal and human models, VEGF has an inhibitory effect on the differentiation of progenitor cells into CD4^+^ and CD8^+^ cells [35,36] and increases the expression of PD-L1, cytotoxic T-cell antigen 4 (CTLA-4), T-cell immunoglobulin and mucin domain 3 (TIM-3), and lymphocyte activation gene 3 protein (LAG3) on the surface of murine T cells [37], which are responsible for the inhibition of effector functions and depletion of T cells. VEGF-stimulated angiogenesis also increases the influx of cells that have an immunosuppressive effect, such as regulatory T cells (Tregs), myeloid-derived suppressor cells and TAMs, into the tumor microenvironment in animal and human models [38,39,40].

High levels of lactate can also inhibit dendritic cell differentiation and maturation [41]. Moreover, a lactate-generated acidic microenvironment promotes the differentiation of neutrophils towards the tumor-promoting N2 profile via the suppressed production of reactive oxygen species (ROS) and reduced phagocytosis, and delay neutrophil apoptosis [42]. In addition, the low pH itself also contributes to local immunosuppression by increased apoptosis as a result of damage to the mitochondrial enzymes of natural killer (NK) cells, which do not have the ability to change their intracellular pH when exposed to an acidic environment. An acidic environment also inhibits the activity of NK cells and decreases the secretion of perforin, granzymes, and cytokines [43], as well as increasing the activity of Tregs, which reduce the anticancer immune response [44].

### 2.3. Other Effects of Lactate in Shaping the Microenvironment

Lactate is also produced by CAFs, which are the second greatest source of this substrate in the tumor microenvironment, after cancer cells themselves [45]. The lactate produced by CAFs is a result of glycolysis stimulated mainly by ROS found in the tumor microenvironment, in contrast to other cells where glycolysis is a consequence of a hypoxic state and acidic pH. The presence of the MCT-1 transporter in cancer cells makes it possible for the lactate produced by CAFs to enter these cells. This relationship between CAFs and tumor cells is described as the inverse Warburg effect [46], because the Warburg effect describes the use of lactate by cancer cells, but here the lactate comes from glycolysis occurring in CAFs.

The inverted Warburg effect allows tumors to become independent of glucose uptake from the microenvironment, which was observed in prostate cancer cells [47].

Lactate production and an acidic environment contribute to hyaluronic acid (HA) production by CAFs [48], which is associated with resistance to treatment and a worse prognosis. Hyaluronian makes the microenvironment more favorable for cell migration by causing accumulation of water and activating the CD44 and RHAMM (receptor for hyaluronian-mediated motility) on the cell surface. Moreover, hyaluronian prevents interaction between ligands of the immune system and receptors on the surface of cancer cells. It also contributes to the accumulation and retention of growth factors such as TGF-β in the microenvironment.

An additional factor related to lactate and its potential impact on angiogenesis is the phosphorylation of tyrosine residues on lactate dehydrogenase A (LDHA), causing a conformational change that increases activity and is more favorable for NADH binding, intensifying the Warburg effect [49,50,51,52].

Lactate is also postulated to have an effect on lysine residues in histones, inducing a similar effect to acetylation and gene activation, which occurs on *VEGF-A* in M2-TAM cells, for example [53]. This is a relatively new discovery, which is currently considered to result from an association between lactate and histone activation. This phenomenon may involve an enzyme or a series of enzymes that would preferentially convert lactate to acetyl residues in TAMs and thus affect tumor angiogenesis. To date, no enzymes have been identified that, as in the case of acetyl residues, would mediate the binding of lactate to histone lysine residues.

## 3. Glutamine

Glutamine has not yet been characterized as an oncometabolite that promotes tumor angiogenesis, either directly or by activating macrophages or fibroblasts. However, its role in tumor angiogenesis becomes important when considered in the context of fumarate and succinate—intermediates in the TCA cycle that stimulate angiogenesis.

Glutamine is a donor of intermediates for the production of lipids, amino acids, hexoamines (precursors for glycosylation of signaling proteins), and glutathione—the main scavenger of ROS in cells [54]. Exogenous glutamine enters cells through the SLC1A5 membrane transporter [55], the expression of which is increased in renal clear cell carcinoma. Once inside the cell, glutamine can be converted in the mitochondria by glutaminase 1 (GLS1) or GLS2 to glutamate and ammonia. GLS1 was found to be a key enzyme associated with the tumor growth of non-small cell lung cancer lines, which have a high rate glutaminolysis [56]. The enhanced expression of GLS 2 was observed in human cervical cancer cells, leading to radio-resistance, contrary to GLS2-silenced cells [57]. The elevated levels of GAC mRNA (Glutaminase C, a splice variant of Kidney type of GLS1 (KGA)) was also observed in gliomas, colorectal cancers and adenomas, and breast cancer cells [58]. Glutamate is then converted to α-ketoglutarate and incorporated into the TCA cycle or carboxylated to isocitrate, which can also enter the TCA cycle. α-ketoglutarate is converted by fumarase in the mitochondria to malonate [59], which is converted into lactate in the cytosol by LDHA. The described glutamine transformations are presented in the diagram below (Figure 3).

HIF changes the TCA cycle from glucose-dependent to glutamate-fed [60] by inhibiting the activity of pyruvate dehydrogenase and pyruvate carboxylase, crucial enzymes in glucose metabolism that obtain energy from this compound. The accumulation of HIF also changes the direction of part of the TCA cycle, which consisted of reactions from citrate to α-ketoglutarate. Normally, in the TCA cycle, citrate bounds with Acetyl-CoA and forms isocitrate. This reaction is catalyzed by citrate synthase. After that, citrate is converted to isocitrate by aconitase. In the next step of the TCA cycle, isocitrate is dehydrogenated to α-ketoglutarate. In tumor cells that use glutamine as a source of energy and building materials, this part of the TCA cycle shows an opposite direction for reactions from α-ketoglutarate to isocitrate. In this manner, α-ketoglutarate, which originates from glutaminolysis, supplies the TCA cycle with the opposite direction. This leads to the accumulation of citrate and other intermediates. As a result, malate and citrate could be produced from isocitrate, which are needed for lipid synthesis, and oxaloacetate, which is used for the biosynthesis of nucleotides in rapidly proliferating cancer cells.

The increased consumption of glutamine has been observed in tumors that have a genetic defect, resulting in a lack of fumarate hydratase and functional succinate dehydrogenase [61,62,63,64]. The lack of activity of these enzymes leads to the accumulation of fumarate and succinate, which promote angiogenesis. In this case, glutamine could be utilized in the TCA cycle in the correct direction, contributing to the accumulation of oncometabolites.

HIF causes the TCA cycle to perform two functions: to provide large amounts of energy for rapidly dividing cells when fed with glucose, and to generate building material when fed with glutamate. Glutamate may also play an additional role in neoplastic cells, as glutamine can support the TCA cycle and maintain a similar metabolic potential to that of normal cells, meaning that most of the initial amount of available glucose can be used to adapt to the microenvironment. This includes the indirect stabilization of HIF by lactate, the stimulation of angiogenesis, and, above all, the maintenance of the necessary acid-base balance in the microenvironment for the proper functioning of enzymes and membrane transporters in tumor and associated cells.

## 4. Succinate

Succinate is a molecule formed from succinyl-CoA by succinyl-CoA synthetase and subsequently converted by succinate dehydrogenase (SDH) to fumarate in the TCA cycle. Under hypoxia [65] or a high H^+^ concentration over the inner membrane of mitochondria [65], succinate is accumulated in the cell as a result of SDH inhibition. High succinate levels can also result from SDH inhibition by hypermethylation or the mutation of SDH subunits [66]. Another pathway to increased succinate concentrations is when the TCA cycle runs in the opposite direction in low-oxygen states [67]. Tumor necrosis factor-associated protein (TRAP1), which inhibits SDH, can lead to increased succinate [68]. In normal tissues, succinate is located in cells and used to generate ATP. Under the pathological conditions in which succinate accumulates, the metabolite is detected in extracellular spaces and acts as a signaling molecule. Moreover, SDH mutations can be found in cancers, including hereditary paraganglioma and pheochromocytoma [69]. The pathogenesis of neuroblastomas, as well as gastrointestinal cancer, colon, renal, and ovarian cancers, might also be connected with succinate [69]. A summary of succinate-modified pathways is presented in Table 1.

### 4.1. Effects of Succinate on Angiogenesis

Succinate exerts a documented effect on angiogenesis via a few pathways (Figure 1). The oncometabolite is associated with the accumulation of HIF-1α due to the inhibition of PHDs, which are responsible for the hydroxylation of HIF-1α and protect it from proteasomal degradation [61,70]. Additionally, increased concentrations of mitochondrial reactive forms of oxygen (mtROS), which, like pure succinate, have the ability to inhibit PHDs, are responsible for HIF-1α accumulation and the stimulation of angiogenesis in SDH-deficient cells [71,72].

Moreover, succinate inhibits ten-eleven translocation proteins (TETs), which sequentially convert the 5-methylcytosine (5-mC) in DNA to 5-hydroxymethylcytosine (5-hmC), 5-formylcytocine, and 5-carboxylcytocine, leading to DNA demethylation [73] and a simultaneous transcriptional increase in both HIF-1α and HIF-2α.

The association between succinate and angiogenesis does not only rely on HIF-1α and HIF-2α. Succinate has also been shown to increase activation of the phosphatidylinositol 3-phosphate kinase (PI3K) pathway, which increases angiogenesis by HIF-1 α-dependent mechanisms, as well as by modulation of the expression of nitric oxide (NO) and angiopoietins [74,75]. Moreover, succinate induces VEGF production through the activation of *STAT3*, a direct transcriptional activator of *VEGF*. Succinate also has the ability to activate the ERK pathway, which promotes sprouting and increases vessel length in tumors [76,77]. Succinate can also promote chemotactic motility, tube-like structure formation, umbilical vascular endothelial cell proliferation, and blood vessel formation [76].

### 4.2. Effect of Succinate on Immune System Cells

Succinate accumulation also affects the immune cells located in the tumor microenvironment and might be involved in lymphocyte-mediated immunity, the regulation of cytokine production, and the immune response-regulation signaling pathway (Figure 2). Investigators found an association between the expression of succinate receptor 1 (GPR91) and the phenotyping of ovarian cancer. GPR91 expression was significantly associated with infiltration by activated CD8^+^ T cells, effector memory CD8^+^ T cells, activated CD4^+^ T cells, effector memory CD4^+^ T cells, Tregs, NK cells, neutrophils, macrophages, activated dendritic cells and B cells, and myeloid-derived suppressor cells in ovarian cancer [29,78]. In addition, an association was found between GPR91 and phenotypic markers of exhausted T cells such as CD244, ENTPD1, CTLA-4, LAG3, and PD1. GPR91 expression also correlated with that of other receptors for inhibitory pathways, such as PD-L1, PD-L2, CD48, CD80, and CD86. GPR91 is not only highly expressed in ovarian cancer cells, so we may extrapolate that succinate exerts similar immune effects in other cancers, such as head and neck squamous cell carcinoma [79]. A recent study [76] also shows that succinate stimulates TAM marker gene expression, including *Arg1*, *Fizz1*, *Mgl1*, and *Mgl2, and is responsible for the upregulation of vascular cell adhesion molecule 1 (VCAM1) and CD11c on the surface of TAMs. Moreover,* succinate was reported to promote TAM polarization through upregulation of HIF-1α and takes part in the induction of IL-6 secretion to the tumor microenvironment, which was pivotal for cell migration in an in vitro model [80]. A recent study by Harber et al. found that cell-permeable diethyl succinate elicits an anti-inflammatory response through reducing tumor necrosis factor (TNF), IL-6, and NO secretion as well as *IL1B* expression at the mRNA level indicates polarization towards VCAM1^+^CD11c^+^CD11b^low^- M2 macrophages known as anti-inflammatory and stimulate angiogenesis and metastases [80]. However, in previous studies, succinate was described as a pro-inflammatory metabolite due to increased IL-1β production mediated by HIF-1α [81,82], so further studies are needed in this area. Macrophage activation can also result from the succinate-induced upregulation of p38 mitogen-activated protein kinases (MAPK), AKT, and AMP-activated protein kinase (AMPK).

Succinate might also stimulate immune cells to suppress host defenses against cancer. Investigators found that immature dendritic cells have a high expression of GPR91, which can be used to enhance their immunostimulatory potential through the induction of chemotactic and pro-inflammatory cytokine production. Additionally, succinate enhances dendritic cell-mediated T-cell activation. After dendritic cell activation, the expression of GPR91 disappears [80].

### 4.3. The Role of Succinate in Neoplastic Stroma

In 2019, Wu et al. [76] reported that succinate promotes cancer metastasis via SUCNR1 (GPR91) by inducing HIF-1α-mediated epithelial-to-mesenchymal transition via PI3K/AKT signaling. The authors found the phosphorylation of p38 MAPK, AKT, and AMPK, which play critical roles in cancer progression. They confirmed that succinate increased HIF-1α protein and mRNA expression with an intensity dependent on the incubation time. An important effect of succinate accumulation is the hypermethylation of histones and DNA cytosine [67,83] as a result of the succinate-mediated inhibition of histone lysine demethylases (KDMs) and TETs [74,84]. Succinate-mediate hypermethylation changes the expression profile of genes, leading to the activation of epithelial-to-mesenchymal transition. Moreover, a recent study showed that succinate suppressed the expression of E-cadherin and raised the expression of N-cadherin and vimentin as well as increasing expression of the epithelial-to-mesenchymal transition transcription factor *SNAIL* at the mRNA level [74]. Data also suggest that cancer-cell-secreted succinate promotes cancer cell migration and invasion through an epithelial-to-mesenchymal transition-dependent mechanism [85].

## 5. Fumarate

Fumarate is another important metabolite of the TCA cycle, formed from succinate and transformed to malate by fumarate hydratase (FH). There is evidence to suggest that inborn and acquired FH dysfunction could be involved in the pathogenesis of renal cell, breast, bladder, and testicular (Leydig cell) cancers, as well as pheochromocytomas, paragangliomas, adrenocortical carcinoma, brain tumors, and sarcoma [86,87].

The loss of function of FH is associated with multiple compensatory metabolic changes, mitochondrial impairment, and the intracellular accumulation of fumarate, as well as an increased sensitivity to DNA damage, which is associated with pseudohypoxia and tumor aggressiveness [10,88]. Compensation for the loss of mitochondrial function is associated with increased aerobic glycolytic rates and higher lactate production, supported by the transcriptional reprogramming of glycolytic enzymes and inhibition of pyruvate dehydrogenase [87]. Sufficient NADH generation in FH-deficient cells is, in turn, maintained by glutamine oxidation [89]. FH-deficient cells also require a constant supply of arginine to ensure the activity of the buffering system that compensates for the potentially toxic accumulation of fumarate [87]. Fumarate accumulation also causes increased levels of ROS in the tumor, which induce lactate production in CAFs despite the upregulation of antioxidant genes [90]. A summary of fumarate-modified pathways is presented in Table 1.

### 5.1. Effect of Fumarate on Angiogenesis

An aberrant accumulation of fumarate contributes to the accumulation of HIFs such as HIF1α and HIF2α and to the induction of angiogenesis via HIF-targeted genes such as *VEGF* and glucose transporter type 1 (*GLUT1*) [10,91]. Fumarate inhibits PHDs, resulting in HIF-1α and HIF-2α stabilization and accumulation [13]. Fumarate also promotes the expression of *HIF-1α* mRNA through tank-binding kinase 1 (TBK1) and the activation of p65 via an NF-κB-dependent pathway [92,93]. Moreover, fumarate has also been shown to increase the expression of *VEGFA* and BCL2 interacting protein 3 *(BNIP3)* mRNAs, directly stimulating angiogenesis [71]. Fumarate also induces V-abl Abelson murine leukemia viral oncogene homolog 1 (ABL1) activity, which upregulates aerobic glycolysis as well as cell proliferation, cell migration, and apoptosis via the mTOR/HIF-1α pathway [94]. However, some data suggest that fumarate, similarly to succinate, reduces levels of *TET1* and *TET2* mRNA, leading to DNA demethylation and a reduction in global 5-hmC levels, resulting in decreased expression of some HIF target genes [61]. Furthermore, HIF-1α accumulation has been shown to be a consequence of ROS accumulation secondary to increased levels of fumarate in mice [95].

### 5.2. Fumarate and Immune System Modification

Data about the immunological consequences of FH depletion in cancer are very limited; however, data on the association between fumarate and the immune system in multiple sclerosis and psoriasis indicate that fumarate can also be viewed as an inflammatory regulator. Dimethyl fumarate (DMF) administered in patients with multiple sclerosis and psoriasis has been shown to downregulate T-cell and B-cell responses through various mechanisms. DMF treatment is associated with the induction of apoptosis in T cells and in monocyte-derived dendritic cells, selective stimulation of T helper 2 cell cytokines, upregulation of Treg cells, and inhibition of migration into injured tissues. DMF also inhibits dendritic cell maturation and modulates antigen presentation abilities, as well as augmenting the cytotoxicity of NK cells, another important component of the innate immune system [93,94,96]. Neutrophils are also affected by DMF through inhibition of the formation and migration of extracellular traps and by suppression of their activity and phagocytosis on the molecular level [97,98]. The aforementioned data on the association between high fumarate concentration and immune system suggest that a similar phenomenon could take place in tumor where fumarate concentration is also higher.

There are no detailed data describing the immunological compartments of tumor microenvironments in FH-deficient cells; however, a recently published study analyzed tumor-infiltrating lymphocytes in cancer patients with FH deficiency. The authors showed that most tumor-infiltrating lymphocytes were CD3-positive T cells. CD8 immunostaining showed 5–30% staining of tumor-infiltrating lymphocytes in 46% of cases; in the remaining cases, 40–90% of tumor-infiltrating lymphocytes were positive for CD8. FoxP3 was positive in only 1–5% of tumor-infiltrating lymphocytes in 77% of cases; in the remaining cases, FoxP3 was positive in 10–50% of tumor-infiltrating lymphocytes [98]. Alaghehbandan et al. analyzed the expression of PD-1/PD-L1 in patients with FH-deficient renal cell carcinoma. Most of the evaluated cases were positive for PD-L1 in tumor cells by qPCR and nine of 13 were positive by immunohistochemistry [99]. Analyses performed by Mitsuko et al. in hereditary leiomyomatosis and renal cell cancer (HLRCC)-associated-renal cell carcinoma with FH deficiency supported the findings reported by Alaghehbandan et al. Most of the examined tumors demonstrated a high expression of PD-L1; however, 21% of specimens showed negative staining. Moreover, the authors detected that PD-L1 expression levels could differ between two separate tumors within the same patient [100]. PD-L1 expression was also previously assessed in papillary renal cell carcinoma, as one of the symptoms of HLRCC syndrome is associated with FH mutation. In a study by Choueiri et al., 10% of papillary renal cell carcinomas were positive for PD-L1 [101]. Furthermore, Motoshina et al. showed that PD-L1 was expressed in 29% of papillary renal cell carcinomas [102]. The authors found no statistically significant differences in PD-L1 expression between type 1 and type 2 papillary renal cell carcinomas (22% vs. 36%). These results shoved a lower expression of PD-L1 than in the studies by Alaghehbandan et al. and Mitsuko et al.; however, a recent comprehensive molecular analysis of papillary renal cell carcinomas demonstrated that type 2 papillary renal cell carcinomas consist of multiple distinct subgroups differing in molecular and phenotypic features, characterized by *CDKN2A* silencing, *SETD2* mutations, *TFE3* fusions, and increased expression of the NRF2-ARE pathway, whereas type 1 papillary renal cell carcinoma is associated with *MET* alterations [103].

### 5.3. DNA Damage Response in FH-Deficient Cells

Fumarate accumulation caused by FH dysfunction makes eukaryotic cells more sensitive to DNA damage [88]. Jiang et al. detected that FH participates in phosphorylation of the histone subtype H2A (H2AX) and checkpoint kinase 2 (CHK2) as well as cell-cycle checkpoint activation, which is essential in the maintenance of the DNA double-strand break response machinery. Upon DNA damage, FH has been shown to translocate to the nucleus, where it produces a local pool of fumarate and inhibits histone H3K36 demethylation and the binding of pro-non-homologous end-joining proteins, which is an important step in the DNA damage response [104]. FH knockdown and high concentrations of fumarate are associated with increased endogenous DNA damage, and increased sensitivity to DNA double-strand breaks [88,105]. It has been shown that a high fumarate level correlates with increased sensitivity to poly-ADP ribose polymerase (PARP) inhibitors [106]. The increased number of dying cells due to impaired DNA repair in FH-deficient cells is probably associated with innate and adaptive immune responses, but more research is needed to assess the association between metabolic disturbances in cancer cells and DNA damage response.

### 5.4. Other Effects of Fumarate in Shaping the Microenvironment

In vitro analyses show that the fumarate-driven inhibition of TET can also suppress microRNA family members called MIR-200, known inhibitors of epithelial-to-mesenchymal transition (EMT). The downregulation of MIR-200could be connected with increased expression of the *Zeb1* and *Zeb2* transcription factors that play an important role in the MET process [94,107]. Investigators also found that FH loss resulted in the hypermethylation and suppression of miR-200 as well as the inhibition of TET activity in HLRCC tissue samples compared to adjacent normal tissue [108]. These results indicate that the promotion of epithelial-to-mesenchymal transition is an additional response to the metabolic disturbance that could induce tumorigenesis.
biology-11-00270-t001_Table 1Table 1Summary of the most important modified pathways.Lactate-modified pathwaysNF-κB/IL-8 pathway in endothelial cells.ERK1/2 and STAT3 pathway.Conversion of lactate to pyruvate, leading to disturbed HIF-1α ubiquitination and accumulation in the cell.Stabilization of HIF-2α within TAMs by activation of mTORC1.Fumarate-modified pathwaysPHD inhibition and HIF-1α and HIF-2α stabilization and accumulation.Inhibition of TETs, leading to DNA demethylation.Accumulation of HIF-1α and HIF-2α and induction of angiogenesis via activation HIF-targeted genes such as *VEGF* and *GLUT1.*Promotion of *HIF-1α* mRNA expression through TBK1 and p65 activation in an NF-κB-dependent pathway.Increased expression of *VEGFA* and *BNIP3.*Induction of proliferation, cell migration, and apoptosis via the mTOR/HIF-1α pathway.Inhibition of the formation of extracellular traps and neutrophils migration; suppression of neutrophil activity and phagocytosis by PI3K/AKT, p38 MAPK, and ERK signaling pathways.Inhibition of the KDM5 family of histone demethylases, which activates the STING/TBK1/IRF3 pathway and increases levels of chemokines.Succinate-modified pathwaysInhibition of PHDs responsible for hydroxylation of HIF-1α.Inhibition of TETs, leading to DNA demethylation.Increased activation of the PI3K pathway.Activation of *STAT3*—a direct transcriptional activator of *VEGF.*Induction of HIF-1α-mediated epithelial-to-mesenchymal transition via PI3K/AKT signaling.ERK pathway activation, which promotes sprouting and increases vessel length in tumors.Inhibition of histone lysine demethylases.NF-κB—nuclear factor kappa B; IL-8—interleukin 8; ERK—extracellular signal-regulated kinase; STAT—signal transducer and activator of transcription; HIF—hypoxia-induced factor; TAMs—tumor-associated macrophages; mTORC1—mammalian target of rapamycin complex 1; PHD—prolyl hydroxylase; TETs—ten-eleven translocation proteins; VEGF—vascular endothelial growth factor; GLUT1—glucose transporter type 1; TBK1—TANK-binding kinase 1; BNIP3—Bcl2 interacting protein 3; PI3K—phosphatidylinositol 3-kinase; MAPK—mitogen-activated protein kinase; STING—stimulator of interferon genes; IRF3—interferon regulatory factor 3.

## 6. Conclusions

Oncometabolites produced by cancer cells and cancer-associated cells located in the tumor stroma act as intracellular messengers that can induce alterations in gene expression. Investigators have shown that the accumulation of lactate, succinate, fumarate, and glutamine can shape the tumor microenvironment, thereby driving cancer progression. In general, oncometabolite accumulation leads to the propagation of a pseudohypoxic signature, contributing to the induction of angiogenesis and an epithelial-to-mesenchymal phenotypic switch. The described molecules also exert significant effects on immune cell activity. Oncometabolites could enhance the antigen-presenting functions required for optimal T cell activation. However, increasing evidence indicates that they are involved in immunosuppressive polarization and T cell exhaustion.

The available data show that several pathways are affected by oncometabolites. However, our review helps to decipher which mechanisms could play a major role in tumor formation. Our results also indicate pathways that could form the basis for the creation of new treatment options involving oncometabolites in tumorigenesis, such as a reduction in lactate production by LDH blockade, glutaminase inhibition, EMT inhibition or DNA damage inhibition. The involvement of oncometabolites in tumorigenesis affects cell differentiation, motility, and invasiveness, which could be further reflected by clinical outcomes. In the future, targeting oncometabolites could help to improve current treatment standards or represent a novel method for fighting against cancer.

## Figures and Tables

**Figure 1 biology-11-00270-f001:**
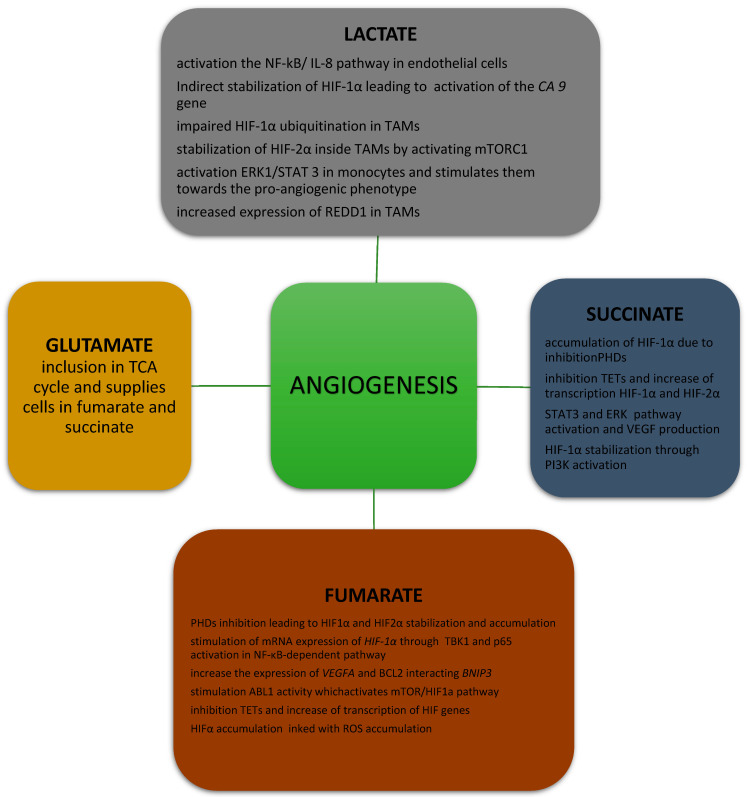
The connection between oncometabolites and angiogenesis. TCA—tricarboxylic acid; NF NF-κB—nuclear factor kappa B; IL-8—interleukin 8; HIF—hypoxia-induced factor; CA9—carbonic anhydrase 9; TAMs—tumor-associated macrophages; mTORC1—mammalian target of rapamycin complex 1; ERK—extracellular signal-regulated kinase; STAT—signal transducer and activator of transcription; REDD1—regulated in development and DNA damage responses 1; PHD—prolyl hydroxylase; TETs—ten-eleven translocation proteins; VEGF—vascular endothelial growth factor; PI3K—phosphatidylinositol 3-kinase; TBK1—TANK-binding kinase 1; BNIP3—Bcl2-interacting protein 3; ROS—reactive oxygen species.

**Figure 2 biology-11-00270-f002:**
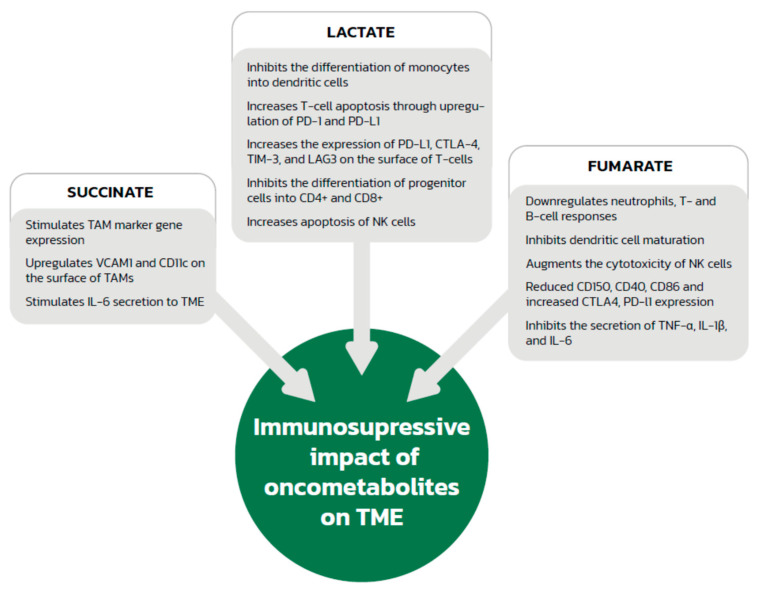
The immunosuppressive effects of oncometabolites. TAMs—tumor-associated macrophages; VCAM1—vascular cell adhesion molecule 1; IL—interleukin; TME—tumor microenvironment; PD1—programmed cell death 1; PD-L1—programmed cell death ligand 1; CTLA-4—cytotoxic T-cell antigen 4; TIM-3—T-cell immunoglobulin and mucin domain 3; LAG3—lymphocyte activation gene 3 protein; NK—natural killer; TNF—tumor necrosis factor.

**Figure 3 biology-11-00270-f003:**
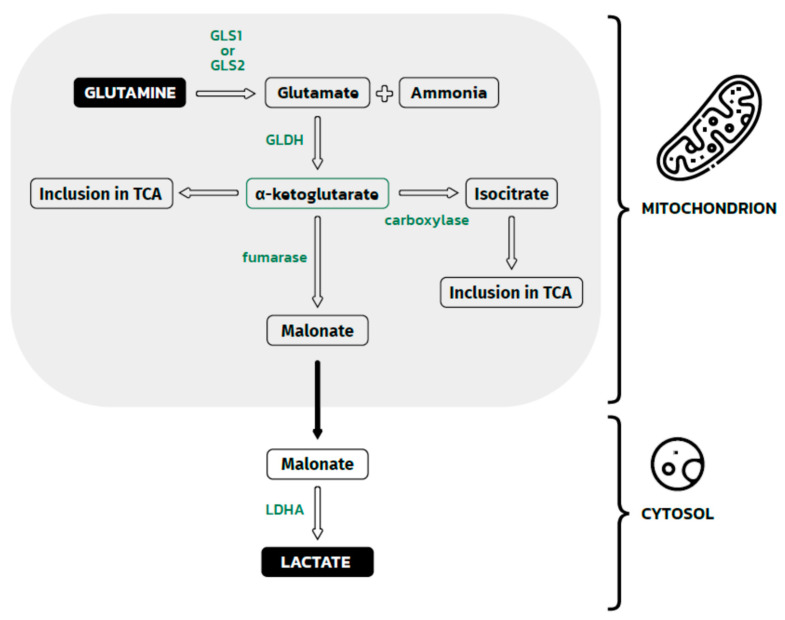
Glutamine metabolism in the mitochondrion and malonate metabolism in the cytosol. The Figure shows glutamine metabolism, which eventually leads to the formation of lactate in the cytosol. This process is dependent on two groups of enzymes: mitochondrial and cytosolic. Inside the mitochondrion, glutamine is deaminated by GLS1 or GLS1 to glutamate and a waste product (ammonia). Then, glutamate is dehydrogenated by GLDH to α-ketoglutarate, which could be directly incorporated into the TCA cycle or indirectly incorporated after carboxylation to isocitrate. Some α-ketoglutarate in mitochondrion is converted to malonate. This molecule crosses the mitochondrial membrane and reaches the cytosolic space. Here, the last pathway step occurs: malate dehydrogenation by LDHA, leading to the formation of lactate. GLS1, GLS2—glutaminase 1, glutaminase 2; GLDH —glutamate dehydrogenase; LDHA—lactate dehydrogenase A; TCA—tricarboxylic acid.

## Data Availability

Not applicable.

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
