# Peer review of "Oncometabolites—A Link between Cancer Cells and Tumor Microenvironment"

_biology, 2022, doi:10.3390/biology11020270_

Round 1

Reviewer 1 Report

Please find all comments depicted within the attached word file.

Reviewer 2 Report

The manuscript is written imprecise. It is has many unsupported statements. One important citation (Yong, C., Stewart, G.D. & Frezza, C. Oncometabolites in renal cancer. Nat Rev Nephrol 16, 156–172 (2020). https://doi.org/10.1038/s41581-019-0210-z) is missing. The review which was not cited by the authors covers the topic of the manuscript.

Line 40-41: Switch in glucose metabolism is not a requirement and does not always happen. In general the first lines of the introduction are written as the body and the tumor are two separate entities which is not the case. It is that the increasing changes in the metabolism of the mutated cells are influencing the microenvironment and the tumor cells are better adapted to the changes.

Line 47: Which important role in the metabolism? What are other molecules that play a role (TNF-alpha etc.)

Line 55-56: The phrase is completely unclear. More explanation about products of anaerobic and aerobic glucose metabolism and its by products is needed.

There are some problems with citations (lines 46, 368).

Reviewer 3 Report

The review of Baryla et al. is adequately informative for both the biological and clinical readership, and in my opinion may be accepted for publication with no major changes. However I would suggest that  the graphics, which appropriately resume the actions of each molecule on the different pathways, are placed altogether at the end of the text. This may facilitate to read  the paper. Moreover, errors of the bibliography should be corrected, as indicated in the text.

Author Response

Dear Reviewer,

Thank you for your attention and careful reading of the manuscript. We have included your opinions and advices in our manuscript and regarding to your suggestion we corrected the list of references. We did not change the order and location of graphics in the text, as we believe that the current location allows the reader to understand the content more easily. In order to refine the manuscript summary, we have changed the location of Table 1, which contains information about important pathways regulated by oncometabolites.

Reviewer 4 Report

The paper is a comprehensive review of the oncometabolytes produced by cancer cells and microenvironment cells, their roles in activating different pathways and potentially stimulating tumor growth.
The paper is well written and structured however several minor issues can be improved.

The introduction part is detailed but lacks references. In particular references should be added for lines 47-56, 65-68 and 69-76.
References should also be added for lines 88-92 and 95-97 in "Lactate" paragraph.

There is also an issue with headings numbering or styling. Paragraphs "Introduction" and "Conclusion" have numbers 1 and 4, however there are no headings with numbers 2 and 3 throughout the paper. Perhaps lines 87, 240, 285 and 371 should be turned into higher ranked headings and headings numbers updated correspondingly.

Several references have a note "error! reference not found" (lines 46, 192, 210, 342, 368). Those references should be checked and corrected to resolve the issue.

To my opinion, the paper can be accepted after addressing mentioned points.

Author Response

Dear Reviewer,

Thank you for your feedbackand careful reading of the manuscript. We have included your advices into our manuscript and have revised the list of references as a result of your suggestion.

Round 2

Reviewer 1 Report

See PDF document

Author Response

Dear Reviewer,

Thank you for your attention and careful reading of the manuscript. We have sent the manuscript to MDPI for English editing and the work has been edited.

Reviewer 2 Report

My suggestions were considered and included in the manuscript. It improved. However, the newly added text has many spelling errors and need to be corrected.  

I disagree with the authors about the Yong et al. paper. As there is a significant overlap in the review by Yong and in the present review. The authors should make more clear where is the difference and just claim "However, in our manuscript, we focused on the analysis of data from original research on solid tumors, and the study published by Yong et al. is a review only for kidney cancer. " is not enough as kidney cancer is a solid tumor and earlier reviews and their conclusion should be considered especially when they partially arrive at the same conclusions.
